# Accurate Determination of the Low-Light-Level Absorption of DUV-Fused Silica at 193 nm with Laser Calorimetry

Fengting Li [1], Haojie Sun [2], Weijing Liu [3], Ruijin Hong [1] and Chunxian Tao [1,*]

1   School of Optoelectronic Information and Computer Engineering, University of Shanghai for Science and Technology, Shanghai 200093, China; 2335054715@st.usst.edu.cn (F.L.); rjhong@usst.edu.cn (R.H.)
2   Department of Printing and Packaging Engineering, Shanghai Publishing and Printing College, Shanghai 200093, China; sunhaojie@sppc.edu.cn
3   Institute of Optics and Electronics, Chinese Academy of Sciences, Chengdu 610209, China; liuweijing@ioe.ac.cn
*   Correspondence: tao@usst.edu.cn

**Abstract:** The low-light-level absorption coefficient of OH-contained and $H_2$-impregnated synthetic fused silica material in 193 nm optical lithography application is determined via a laser calorimetry measurement. The fluence and repetition rate dependences of the absorptances of the deep ultraviolet (DUV)-fused silica samples with different thickness are measured. The measured dependences are fitted to a theoretical model, taking into consideration the generation and annealing of laser irradiation induced defects. The surface absorption, the low-light-level linear absorption coefficient, as well as the nonlinear absorption coefficient of the fused silica material are accurately determined via the fitting. The low-light-level linear absorption coefficients determined via the fluence dependence and the repetition rate dependence are in good agreement, demonstrating the reliability of the measured low-light-level absorption coefficient, which is the key parameter to the determination of the internal transmission of the DUV-fused silica material used in the 193 nm optical lithography.

**Keywords:** low-light-level absorption; internal transmission; DUV fused silica; laser calorimetry; defects

## 1. Introduction

For ultra-large scale integrated circuit (IC) manufacturing, even though the optical micro-lithographic technique has advanced to EUV [1,2], optical lithography with DUV ArF excimer laser at 193 nm is still the workhorse for high volume chip production [1,3]. In 193 nm optical micro-lithographic systems, the most widely used wide band-gap optical material is synthetic fused silica due to its high transmission, low birefringence, and easy manufacturing processing [4,5]. The internal transmission at 193 nm is the fundamental optical property of the DUV-fused silica material used in 193 nm optical lithographic tools, as a large number (>20) of optical elements with large dimensions (>300 mm in diameter) are needed. In most cases, an internal transmission higher than 99.7% cm$^{-1}$ is required [6] to maximize the overall transmission of the laser beam illuminating the silicon wafer for IC chip production, therefore improving the production efficiency. However, the accurate determination of such a high internal transmission is not an easy task, as it is behind the accuracy (with a typical accuracy of ±0.3%) of a commercial DUV/VUV (Vacuum UV) spectrophotometer. Instead, the internal transmission is usually evaluated via absorption measurement with photometry [7], laser calorimetry (LCA) [8,9], laser induced deflection (LID) [10,11], as well as Hartmann–Shack wavefront sensor (WFS) based photothermal technique [12,13]. Still, the accurate determination of the internal transmission is difficult, as it is related only to the low-light-level linear absorption or called initial absorption [14]. However, in absorption measurement with LCA, LID or WFS in which an ArF excimer laser is used as the irradiation light source, the measured absorption normally includes

the contributions from the surface absorption, the linear absorption, the nonlinear or two-photon absorption, and defect-related absorption [6,15,16]. Some of these contributions are irradiation fluence and repetition rate dependent [17]. To obtain the internal transmission related low-light-level linear absorption, these absorptions have to be separated.

On the other hand, sensitive absorption measurement is beneficial to investigating the long-term transmission degradation effect and lifetime [18] of the DUV-fused silica materials and absorption-induced wavefront distortion [19] inside the projection lenses of the lithography optics. The lifetime is another important characteristic property of the DUV-fused silica material used in 193 nm optical lithography. For IC production a lifetime of up to $10^{11}$ pulses is needed [20]. The measurements of the fluence and repetition rate dependences of the absorption, together with the photoluminescence excited at 193 nm [21,22], provide useful information on the structural defects such as the oxygen deficient center (ODC) [23], the E' center [23], and the non-bridging oxygen hole center (NBOHC) [24], which significantly affect the lifetime. The dependence of the absorption characteristics on the irradiation laser parameters (pulse energy, pulse duration, and repetition rate, etc.) is the key to a better understanding of the long-term degradation effect of the DUV-fused silica material under 193 nm laser irradiation.

In this paper, LCA is employed to measure the absorption behavior and further to determine the internal transmission related low-light-level absorption of DUV-fused silica at 193 nm. By directly fitting the measured fluence dependence of absorption of the fused silica samples with different thickness to an appropriate theoretical model, the surface absorption, linear absorption, and nonlinear absorption are separated. For comparison, the repetition rate dependence of the absorption is also measured and employed to separate the linear and nonlinear absorption. The low-light-level linear absorption coefficient is determined from the separated linear absorption. The determined low-light-level linear absorption coefficient is in good agreement with that reported previously in the literature. Together with the measured scattering loss coefficient, the linear absorption coefficient can be used to evaluate the internal transmission of the DUV-fused silica material.

## 2. Experimental Details

An LCA instrument (Laser Zentrum Hannover, Hannover, Germany) with 193 nm laser irradiation is employed to measure the absorption (absorptance) of the DUV-fused silica according to ISO 11551 [25]. The experimental setup is presented in Figure 1. An ArF excimer laser (Indystar, Coherent, Santa Clara, CA, USA) with a maximum repetition rate 1000 Hz, a pulse duration 10.5 ns, and a beam diameter 2 mm is used to irradiate the DUV-fused silica sample. The pulse energy (irradiation fluence) is adjusted using a variable attenuator and monitored using a photo-detector. The output of the photo-detector is calibrated via an energy meter to give the energy of the laser pulse. The irradiation repetition rate is adjusted via the laser power controller. In LCA, the sample is placed in a thermally isolated chamber to minimize the influence of the environmental temperature fluctuation on the sensitive temperature measurement. The laser irradiation induced temperature rise in the fused silica sample is measured with a negative temperature coefficient (NTC) thermocouple touched to the rear surface of the sample. The temperature sensitivity of the NTC is better than 0.1 mK [9]. The distance between the laser irradiation site and the temperature measurement site is 7 mm [25]. The whole beam path of the setup is sealed. The absorption measurement is performed under high-purity nitrogen purging, with the air in the beam path and sample chamber being totally replaced by the nitrogen. The measured temperature rise is used to determine the absorptance of the sample under irradiation.

In the LCA measurement, once the sample is installed inside the isolation chamber, a waiting period (a half-hour to a couple of hours) is needed for the thermal stabilization of the test chamber before the test. Then, the test is performed in three successive intervals: the drift recording interval of at least 30 s, the heating interval of 5 to 300 s during which the laser beam irradiates the test sample surface, and the cooling interval of at least

200 s immediately after the laser irradiation is stopped. It is expected to finish one LCA measurement in at least half an hour. Figure 2 shows a typical temperature curve of the LCA measurement with a heating interval of 120 s. The absorptance of the test sample is obtained by fitting the measured temperature rise curve to a theoretical model assuming an infinite thermal conductivity of the sample. The temperature rise expression is [25,26]:

$$\Delta T(t) = \begin{cases} 0, & t \leq t_1 \\ \frac{AP}{\gamma C_{\text{eff}}}\left[1 - e^{-\gamma(t-t_1)}\right], & t_1 \leq t \leq t_2 \\ \frac{AP}{\gamma C_{\text{eff}}}\left(e^{\gamma t_2} - e^{\gamma t_1}\right)e^{-\gamma t}, & t \geq t_2 \end{cases} \tag{1}$$

where $A$ is the absorptance of the test sample, $P$ is the laser power (product of pulse energy and repetition rate), and $\gamma$ is the heat loss coefficient. And

$$C_{\text{eff}} = \sum_i m_i c_{\text{p}i} \tag{2}$$

is the effective heat capacity of the test sample and the holder, and $m_i$ and $c_{\text{p}i}$ ($i$ = 1, 2) are the mass and heat capacity of the test sample and the holder, respectively. It is worth mentioning that with LCA the measured absorptance is only related to the absorbed energy which is converted to heat via non-radiative relaxation. The absorbed energy which is converted to fluorescence via radiative relaxation cannot be detected by the LCA measurement. For the fused silica material, a fraction of the absorbed energy is converted to photoluminescence. However, it is believed that the fraction of non-thermal relaxation is small compared to that converted to heat [8,10,12]. The influence of the photoluminescence on the absorptance measurement of the fused silica material is therefore neglected.

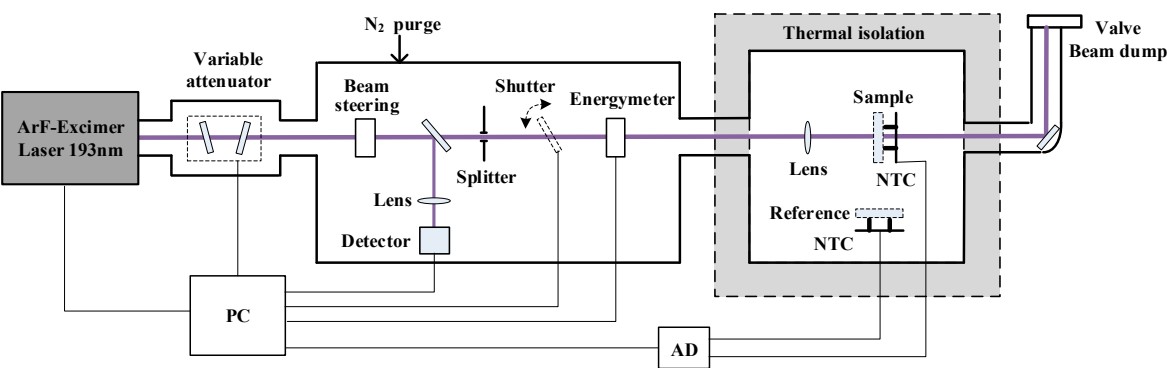

**Figure 1.** Schematic diagram of the LCA experimental setup. NTC: Negative-Temperature-Coefficient thermocouple; AD: A/D converter; PC: Personal computer.

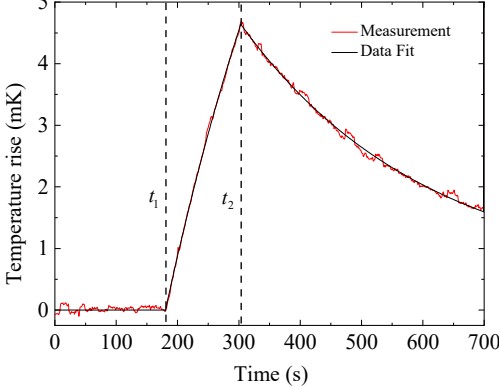

**Figure 2.** Typical temperature curve for LCA measurement.

In the experiment, the absorptance of the DUV-fused silica is measured under different fluence and repetition rates. From the fluence and repetition rate dependences, the absorption channels (surface, linear, and nonlinear absorptions) are separated, and the low-light-level absorption related to the internal transmission is determined.

The fused silica material investigated is intended for 193 nm lithography application. It is OH-contained and $H_2$-impregnated. The OH and $H_2$ contents are 250 ppm and $2 \times 10^{16}$ cm$^{-3}$, respectively. Four fused silica samples with thicknesses of 2.1 mm, 4.0 mm, 5.7 mm, and 8.1 mm are prepared from the same fused silica rod (with diameter 25.4 mm and length 100 mm) and polished with the same mechanical polishing process. Therefore these samples are assumed to have the same surface absorption and bulk absorption (both linear and nonlinear absorption). During the LCA measurement, the laser irradiation induced conditioning effect and degradation effect, as presented in Figure 3, should be accounted for to avoid additional measurement errors. Figure 3a shows the dependence of the transmission on the irradiation dose with a fluence 22 mJ/cm$^2$ and a repetition rate 1000 Hz, showing the transmission degradation. The transmission first increases due to the laser conditioning effect and then decreases with the increasing irradiation dose. This transmission drop is permanent and non-reversible. The transmission is not recovered after stop irradiation for 24 h, as possible recovery could only be the annealing of E' centers by the impregnated $H_2$ which takes place in tens of seconds after stopping laser irradiation [27]. Correspondingly, Figure 3b shows the dependence of the absorptance on the irradiation dose with a fluence 10 mJ/cm$^2$ and a repetition rate 1000 Hz. Again, the absorptance first decreases due to the laser conditioning effect, and then increases with the increasing dose due to laser-induced degradation. Therefore, to determine accurately the absorption behavior of the fused silica sample, a pre-irradiation is needed and the irradiation dose has to be optimized to eliminate the influence of the surface contamination (laser conditioning effect) while minimizing the degradation effect. Figure 4 shows a typical laser conditioning effect of the 5.7 mm-thickness fused silica sample. From the data, the measured absorptance stabilizes after approximately 4 kJ/cm$^2$ irradiation dose. Therefore, before the LCA measurement, the fused silica sample is pre-irradiated with 4 kJ/cm$^2$ dose at a relatively low laser fluence (below 5 mJ/cm$^2$). Then, the absorptances of fused silica samples with different thickness are measured as a function of the laser fluence and of the repetition rate. From the measured fluence and repetition rate dependences, the absorption channels are separated and the low-light-level linear absorption coefficient is determined.

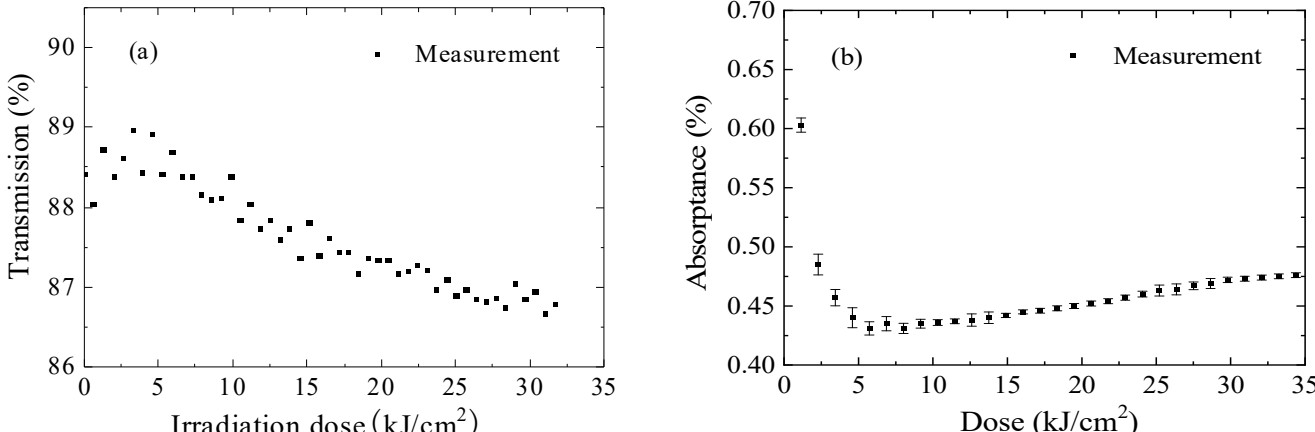

**Figure 3.** Degradation effect of DUV-fused silica material under 193 nm laser irradiation. (**a**) Transmission degradation at fluence 22 mJ/cm$^2$, 1000 Hz repetition rate; (**b**) absorptance behavior at fluence 10 mJ/cm$^2$, showing laser conditioning effect and laser-induced degradation (absorption increasing).

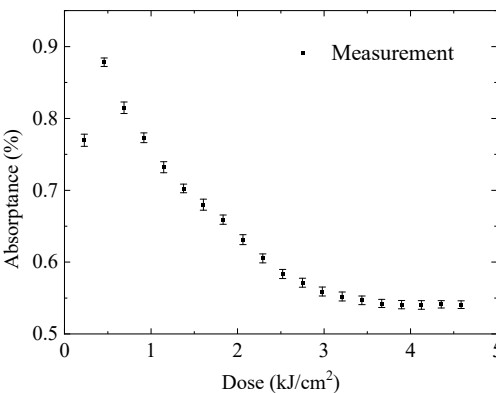

**Figure 4.** Laser conditioning effect of the 5.7-mm-thick fused silica sample.

## 3. Results and Discussion

For OH-contained and $H_2$-impregnated fused silica material for 193 nm lithography application, there are several structural defects which could be related to the absorption behavior at 193 nm. Figure 5 shows the photoluminescence (PL) spectra of the fused silica sample excited at 193 nm. Fluorescence bands related to ODC (290 nm and 320 nm bands), peroxide radical (POR) (390 nm band), and NBOHC (650 nm band), as well as a 550 nm band of unknown defects which is believed to be related to the strained bonds [28], are present. In addition, E' centers, which have no 193 nm-excited fluorescence band in the 200~800 nm spectral range, are believed to be the major source to the 193 nm absorption [17]. In the DUV fused silica material the E' centers are produced by the 193 nm laser irradiation induced breaking of the strained Si-O-Si bonds [17,29] and annealed by the impregnated $H_2$ [30].

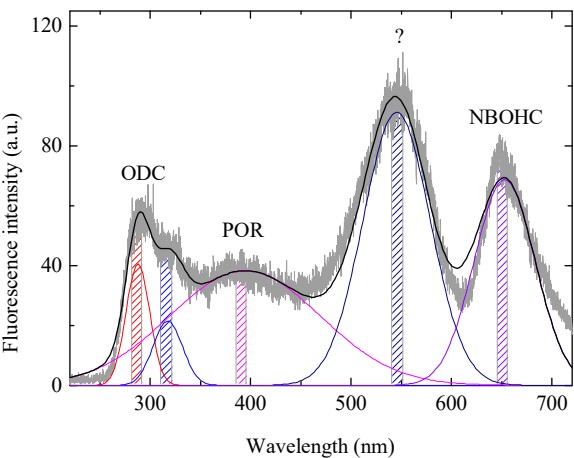

**Figure 5.** The photoluminescence spectrum of the fused silica sample excited at 193 nm. The black line represent the best fit to the measurement. The color lines represent the best fits to PL bands related to different defects.

Taking into consideration the surface absorption, the intrinsic and defect-related linear and nonlinear absorption, as well as the generation and annealing of the absorption-related defects, the absorptance of the DUV-fused silica sample measured by the LCA instrument can be expressed as [17].

$$A(H, f = \text{const}) = A_S + \alpha_0 \times d + \frac{c_1 \cdot H + c_2 \cdot H^2}{1 + c_3 \cdot H} = A_0 + \frac{c_1 \cdot H + c_2 \cdot H^2}{1 + c_3 \cdot H} \qquad (3)$$

when the repetition rate $f$ keeps constant and laser fluence $H$ changes. And,

$$A(f, H = \mathrm{const}) = A_S + \left( \alpha_0 + \frac{\beta_0}{\tau} \times H + \frac{c_1^* \cdot f}{c_2^* + c_3^* \cdot f} \right) \times d \qquad (4)$$

when the laser fluence $H$ keeps constant and the repetition rate $f$ changes. In Equations (3) and (4), $As$ is the surface absorption, $d$ is the sample thickness, and $\alpha_0$ and $\beta_0$ are the linear and nonlinear absorption coefficients, respectively, $\tau$ is the pulse duration. $A_0 = A_S + \alpha_0 * d$ is the low-light-level absorption with $H = 0$. $c_1$, $c_2$, $c_3$, $c_1^*$, $c_2^*$, and $c_3^*$ are constants related to the optical properties and thickness of the fused silica sample and to the fluence and repetition rate of the laser pulse. These constants are to be determined by fitting the measured laser fluence and repetition rate dependences of the absorptance of the fused silica samples to Equations (3) and (4), respectively. Previous measurement results showed that the surface absorption has no observable contribution to the measured nonlinear absorption of the fused silica, it is believed to be a linear effect [31].

Figure 6 shows the laser fluence dependence of the absorptance of the DUV-fused silica. The dependences of the absorptance of the fused silica samples with thickness 2.1 mm, 4.0 mm, 5.7 mm, and 8.1 mm are presented in Figure 6a. The LCA measurements are performed with a repetition rate of 1000 Hz. The fluence range is 0.1 to below 4 mJ/cm² to minimize degradation. Due to the sensitivity limit of the LCA instrument, the minimum detectable absorptance is approximately 0.2%. As expected, the absorptance increases nonlinearly with the laser fluence. At the low energy (low-light-level) end, the absorption is dominated by the surface absorption and linear absorption ($A_0$ in Equation (3)). As the pulse energy increases, the absorption increases rapidly due to the additional absorption of the irradiation induced E' centers and other structural defects. A strong nonlinear dependence of the absorption on the pulse energy is observed in the fluence range 0~2.5 mJ/cm². Above approximately 2.5 mJ/cm², the contribution of the nonlinear (two-photon) absorption becomes apparent. The fluence dependence of the absorption gradually becomes linear [17,31]. By setting $A_0$, $c_1$, $c_2$, and $c_3$ in Equation (3) as free parameters and fitting the measured laser fluence dependence of the absorptance to Equation (3) via multi-parameter fitting, the $A_0$ value for each fused silica sample is determined, and its dependence on the sample thickness is presented in Figure 6b for the four samples. A good linear relation between $A_0$ and the sample thickness is obtained. A linear fit gives a surface absorption of 0.079% and a linear absorption coefficient of $2.10 \times 10^{-3}$ cm$^{-1}$ for the DUV-fused silica material. By measuring the absorptances of the fused silica samples with different thickness, the surface absorption and bulk linear absorption are well separated.

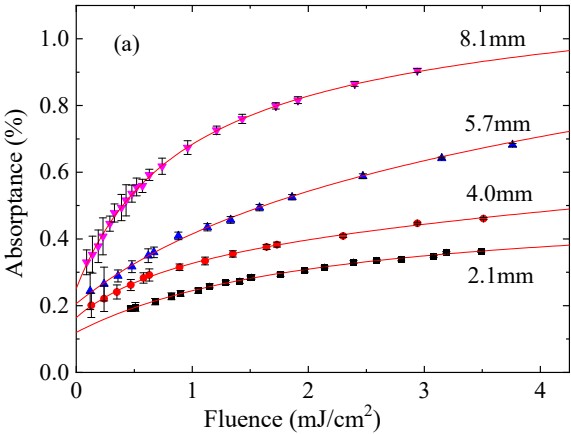
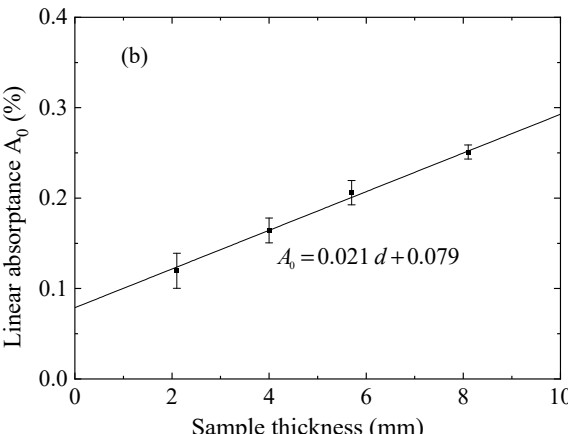

**Figure 6.** (**a**) The fluence dependence of absorptance of fused silica sample with different thickness and corresponding best fits; (**b**) the linear absorptance versus sample thickness showing the separation of surface absorption and bulk linear absorption. Symbols: measurements; lines: best fits.

Once the surface absorption is determined, Equation (3) can be re-written in the form of absorption coefficient to the following:

$$\frac{A(H, f = \text{const}) - A_S}{d} = \alpha_0 + \frac{c_1 \cdot H + c_2 \cdot H^2}{(1 + c_3 \cdot H) \cdot d} \tag{5}$$

The data presented in Figure 6a is re-drawn in terms of absorption coefficient by eliminating the surface absorption and is shown in Figure 7. There is a significant deviation between the absorption coefficients calculated from the absorptance values measured with the 2.1 mm-thickness fused silica sample and from the absorptance measured with the samples with 4.0 mm, 5.7 mm, and 8.1 mm thickness. Again, by setting $\alpha_0$, $c_1$, $c_2$, and $c_3$ in Equation (5) as free parameters and fitting the measured laser fluence dependence of the calculated absorption coefficient to Equation (5) via multi-parameter fitting for each sample, the $\alpha_0$, $c_1$, $c_2$, and $c_3$ values for each fused silica sample are obtained. The results are summarized in Table 1.

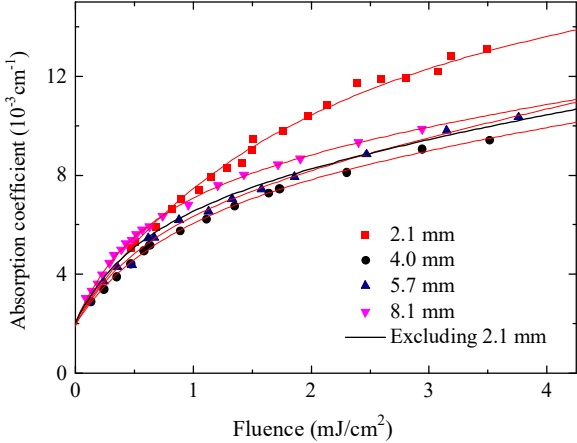

**Figure 7.** The fluence dependence of absorption coefficient of the fused silica sample with different thickness and corresponding best fits. Symbols: measurements; lines: best fits.

**Table 1.** $\alpha_0$, $c_1$, $c_2$ and $c_3$ values for the DUV fused silica samples with different thickness.

| Fit Parameter | Sample Thickness (mm) | | | | Average | Excluding 2.1 mm |
| --- | --- | --- | --- | --- | --- | --- |
| | 2.1 | 4.0 | 5.7 | 8.1 | | |
| $\alpha_0$ ($10^{-3}$ cm$^{-1}$) | 2.07 | 1.97 | 1.91 | 2.00 | 1.99 | 2.02 |
| $c_1$ ($10^{-3}$ cm$^2$/mJ) | 7.25 | 7.50 | 9.20 | 11.68 | 8.91 | 10.05 |
| $c_2$ ($10^{-3}$ cm$^4$/mJ$^2$) | 0 | 0.56 | 1.23 | 0.91 | 0.68 | 1.10 |
| $c_3$ ($10^{-3}$ cm$^2$/mJ) | 0.37 | 0.98 | 1.36 | 1.48 | 1.05 | 1.46 |

From the data presented in Table 1, when the laser fluence is higher than 1 mJ/cm$^2$, an absorption coefficient deviation between the 2.1-mm-thick fused silica sample and other three samples is present. This deviation is mainly due to the $c_2 H^2$ and $c_3 H$ terms in Equation (5), which are related to the generation and annealing of the E' centers [17]. This deviation may be caused by neglecting the absorption contribution of defects other than E' centers in describing the absorption behavior with Equation (3). For example, the NBOHC defects, which are co-generated with the E' centers during the 193 nm laser irradiation and are also annealed by the impregnated H$_2$ during the dark periods between laser pulses, have two optical absorption bands in the DUV spectral range peaked at 257 nm (4.8 eV, with full width at half maximum (FWHM) 1.07 eV) and 181 nm (6.8 eV, FWHM 1.76 eV, a combination of multiple bands) [24,32]. On the other hand, when the laser fluence is below 1 mJ/cm$^2$, this deviation becomes negligible, as the absorption behavior of the DUV-fused silica is dominated by the surface absorption and intrinsic linear absorption of

the DUV-fused silica at the low fluence level. Above approximately 2.5 mJ/cm$^2$, the slopes of fluence dependences of the absorption coefficient for the four samples are close, as the slopes are dominated by the nonlinear absorption coefficient of the fused silica material. The determined linear absorption coefficients ($\alpha_0$) with the four DUV-fused silica samples are between 1.91~2.07 $\times$ 10$^{-3}$ cm$^{-1}$, with an average of 1.99 $\times$ 10$^{-3}$ cm$^{-1}$. This value is very close to the value 2.02 $\times$ 10$^{-3}$ cm$^{-1}$, determined by simultaneously fitting the absorption coefficient data of the other three DUV-fused silica samples (excluding the data of the 2.1-mm-thick sample) to Equation (5), as presented in Table 1.

In addition, the repetition rate dependence of the absorptance of the 4.0 mm-thickness DUV-fused silica sample is also measured. Figure 8a shows the absorption coefficient (after eliminating the surface absorption) versus the repetition rate *f*, measured at three different laser fluences and the corresponding best fits to Equation (4). The measurements start from the low repetition rate and end at the high repetition rate. After the measurement of each dependence is finished, the absorption is re-measured at a lower repetition rate (marked by the purple star symbols in Figure 8a) to check the possible occurrence of irradiation-induced degradation. No degradation is observed for the repetition rate dependence measurement. From the fitting curves, the absorption coefficients at $f = 0$ ($\alpha_0 + \beta_0 \times H/\tau$) are extracted for the three fluences and are presented in Figure 8b. From the linear fit of the fluence dependence of the absorption coefficient, the linear absorption coefficient $\alpha_0$ is determined to be 2.05 $\times$ 10$^{-3}$ cm$^{-1}$, while the nonlinear absorption coefficient $\beta_0$ is determined to be 12.5 $\times$ 10$^{-9}$ cm/W. Again, the determined linear absorption coefficient 2.05 $\times$ 10$^{-3}$ cm$^{-1}$ is very close to that determined from the measured fluence dependence of the absorptance, 2.10, 1.99, or 2.02 $\times$ 10$^{-3}$ cm$^{-1}$. The good agreement among the values obtained via different approaches indicates the reliability of the determined low-light-level linear absorption coefficient of the DUV-fused silica material. In addition, the extracted nonlinear absorption coefficient is consistent with the values previously reported [17,31,33], confirming the feasibility of accurate determination of the absorption parameters of highly transparent DUV optical materials.

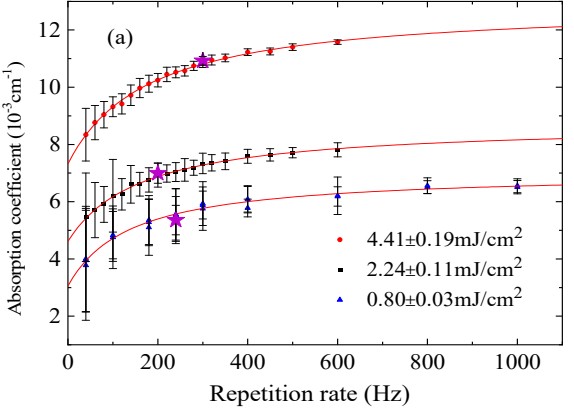 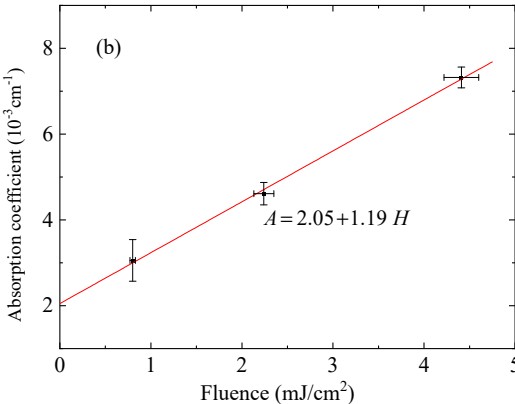

**Figure 8.** (**a**) The repetition rate dependence of absorption coefficient of the 4.0-mm-thick fused silica sample measured at different irradiation fluence and the corresponding best fits; (**b**) bulk absorption coefficient versus irradiation fluence showing the separation of linear and nonlinear absorptions. Symbols: measurements; lines: best fits; Purple star: re-measurement, see text for details.

It is worth mentioning that in this paper the absorption measurement is performed with an industrial excimer laser with a pulse duration ~10.5 ns, not the excimer laser intended for micro-lithography, which has a typical pulse duration up to 160 ns via a pulse stretcher [34,35]. While the linear absorption is independent on the pulse duration, the nonlinear absorption, both intrinsic and induced by irradiation generated defects, is inversely proportional to the pulse duration. Therefore, the contribution of the nonlinear absorption to the measured absorption via LCA is significantly reduced for a long pulse

duration. In 193 nm lithography, the use of a long pulse duration is beneficial to increasing the overall transmission and lifetime of the projection lens.

For the internal transmission determination of the DUV-fused silica, the bulk scattering loss has to be included as the following:

$$T = 1 - \exp[-(\alpha_\text{s} + \alpha_0) \cdot d] \tag{6}$$

where $T$ is the internal transmission and $\alpha_\text{s}$ is the bulk scattering coefficient. For DUV-fused silica material intended for 193 nm lithography application, the bulk scattering coefficient ranges from 0.6 to $1.7 \times 10^{-3}$ cm$^{-1}$, depending on the OH content and fictive temperature [36,37]. By taking into account the bulk scattering, the internal transmission of the investigated DUV-fused silica material is in the range from 2.6 to $3.7 \times 10^{-3}$ cm$^{-1}$, which is acceptable to the application in 193 nm optical lithography.

## 4. Conclusions

In summary, LCA was employed to investigate the absorption characteristics of DUV-fused silica material for the purpose of determining the low-light-level absorption coefficient to estimate the internal transmission, a fundamental parameter for the DUV-fused silica in 193 nm lithography application. The absorptance of DUV-fused silica samples with different thickness were measured as a function of the irradiation fluence and of the repetition rate. The measured fluence and repetition rate dependences of the absorptance were fitted to a theoretical model, taking into consideration the generation and annealing of E′ center defects. The surface absorption and low-light-level linear absorption coefficient of the DUV-fused silica samples were determined via the fluence dependence, and the low-light-level linear absorption coefficient and nonlinear absorption coefficient were obtained via the repetition rate dependence. The low-light-level linear absorption coefficients determined via the fluence dependence and the repetition rate dependence were in good agreement, and were also in consistent with values previously reported in the literature. The good agreement demonstrated the reliability of the measured low-light-level absorption coefficient, which could be used to determine the internal transmission of the DUV-fused silica material used in the 193 nm optical lithography. The absorption measurement method presented in this paper is expected to find applications to the determination of the low internal transmission of other DUV optical materials such as CaF$_2$ [4,38] and to the investigations of the optical degradation of DUV optical materials [20,39].

**Author Contributions:** Conceptualization, R.H.; methodology, W.L.; software, F.L. and H.S.; validation, C.T.; investigation, F.L. and W.L.; resources, R.H. and C.T.; data curation, W.L.; writing—original draft preparation, H.S.; writing—review and editing, C.T.; visualization, F.L. All authors have read and agreed to the published version of the manuscript.

**Funding:** This research received no external funding.

**Institutional Review Board Statement:** Not applicable.

**Informed Consent Statement:** Not applicable.

**Data Availability Statement:** Data of the results presented in this article are available from the corresponding author upon reasonable request.

**Conflicts of Interest:** The authors declare no conflicts of interest.

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
