# Peer review of "Accurate Determination of the Low-Light-Level Absorption of DUV-Fused Silica at 193 nm with Laser Calorimetry"

_photonics, doi:10.3390/photonics11040305_

Round 1

Reviewer 1 Report

Comments and Suggestions for Authors

Comments on the Quality of English Language

The text uses many long sentences, which are sometimes difficult to understand and need to be revised.

Author Response

Comments on the Quality of English Language: The text uses many long sentences, which are sometimes difficult to understand and need to be revised.

Reply: The text of the manuscript is systematically revised for easy understanding. The long sentences are shortened.

The authors Fengting Li et al. measured fluence and repetition rate dependences of the absorptances of the deep ultraviolet (DUV) fused silica samples with different thickness to a theoretical model. The topic of this paper is well chosen because the absorption of silica in the visible and near-infrared bands has been extensively measured. However, the absorption characteristics of silica in the deep ultraviolet band are not well understood. The author obtained the absorption data through a large number of repetitive implementations, and fitted the obtained data with formulas, which required a large workload. Therefore, I agree to accept this article with minor revision. I have two questions that the author needs to answer.

  1. In Figure 8b, the author only gives three test data points to fit a straight line. This is very dangerous. Perhaps limited by the laser energy, we cannot give the absorption coefficient at higher energies, but the author may wish to give a few more test data points at pulse energies less than 5 millijoules. This way the conclusion of the article will be more convincing.

Reply: The reviewer is absolutely right. The problem for us is that to obtain the experimental data for even one data point presented in Fig. 8(b) is extremely time-consuming and expensive, as at least around 10 absorptance measurements were to be performed at different repetition rates in order to get one data point presented in Fig. 8(b), as shown in Fig. 8(a). It took at least half hour to finish one absorptance measurement shown in Fig. 8(a) and took one whole day to finish one measurement of the repetition rate dependence of the absorptance (corresponding to one curve in Fig. 8(a)). Therefore, only a minimum data points were measured, as required to fit a linear line in Fig. 8(b). Since the linear dependence is theoretically predicted, and there is no source (such as the surface absorption) for possible nonlinear dependence, we think three data points for an expected linear fit is acceptable. More experimental details are added in the revised manuscript to imply the time-consuming behavior of the absorptance measurement experiment.  

  1. Can the author do some analysis on the absorption mechanism under different energies? This is because at high energies there are both linear absorption and nonlinear absorption, and sometimes it is difficult to distinguish them.

Reply: Some analysis is added in the revised manuscript to discuss the absorption mechanism under different energy levels. Briefly, at low-energy level, the absorption is dominated by the linear absorption. Above approximately 2.5 mJ/cm2 fluence, the absorption is mainly due to the nonlinear absorption. In between the nonlinear dependence of the absorption on the fluence is caused by the irradiation induced defects.

Reviewer 2 Report

Comments and Suggestions for Authors

The manuscript of F. Li et al. is dedicated to a study of laser induced absorption effects in fused silica at the wavelength 193 nm. Besides the applied laser calorimetric measurement facility, the authors describe the measurement protocol and the annealing procedure of the tested samples. Results include the surface absorption, the linear and nonlinear absorption, and the repetition rate dependence of the absorption.  

The illustrated research work stands in a long row of laser calorimetric investigations in optical materials and contains some results of interest for the scientific community in the field of laser technology and material development. The topic of the paper fits approximately to the scope of the Photonics Special Issue on Optoelectronic Detection Technologies and Applications. In the present form, the manuscript is well structured, written in a comprehensive style and citing numerous references in a correct context. Consequently, the paper is worthwhile a publication in an archival journal. Prior to publication, some minor items may be considered for clarification:

§  The recovery time of 24 h after annealing may be relatively short. The authors may comment on investigations over longer time periods.

§  The calorimetric measurement is performed with laser pulses in the ns-regime. It would be interesting to consider if there are any dependencies of the calorimetric measurement on the pulse duration.

§  The production process for the fused silica samples may be briefly specified. For example, there may be a dependence of the material properties on the position in the ingot.

§  Part of the absorbed laser energy will be converted to fluorescence radiation, which is not detected by the laser calorimetric method. It would be interesting to have an estimation concerning the fraction of this loss channel compared to the absorption value detected by the laser calorimeter.

§  The author may discuss and assess effects of nonlinear surface absorption.

Author Response

The manuscript of F. Li et al. is dedicated to a study of laser induced absorption effects in fused silica at the wavelength 193 nm. Besides the applied laser calorimetric measurement facility, the authors describe the measurement protocol and the annealing procedure of the tested samples. Results include the surface absorption, the linear and nonlinear absorption, and the repetition rate dependence of the absorption.

The illustrated research work stands in a long row of laser calorimetric investigations in optical materials and contains some results of interest for the scientific community in the field of laser technology and material development. The topic of the paper fits approximately to the scope of the Photonics Special Issue on Optoelectronic Detection Technologies and Applications. In the present form, the manuscript is well structured, written in a comprehensive style and citing numerous references in a correct context. Consequently, the paper is worthwhile a publication in an archival journal. Prior to publication, some minor items may be considered for clarification:

  1. The recovery time of 24 h after annealing may be relatively short. The authors may comment on investigations over longer time periods.

Reply: There is no reported long-term annealing mechanism for fused silica. The only possible recovery mechanism is the annealing of E’ centers by the impregnated H2 which takes place in tens of seconds. This statement is added in the revised manuscript and a reference is added to support this statement.

  1. The calorimetric measurement is performed with laser pulses in the ns-regime. It would be interesting to consider if there are any dependencies of the calorimetric measurement on the pulse duration.

Reply: One paragraph is added to discuss the dependence of the measured absorptance on the pulse duration of the excimer laser. In brief, the linear absorption is independent of the pulse duration, but the nonlinear absorption is inversely proportional to the pulse duration. Stretching the pulse duration will reduce the overall absorption and increase the lifetime of the projection lens.

  1. The production process for the fused silica samples may be briefly specified. For example, there may be a dependence of the material properties on the position in the ingot.

Reply: We are sorry that no information on the raw material of the fused silica samples could be provided. The samples were cut from a fused silica rod of diameter 25.4mm and length 100mm and polished via conventional mechanical polishing method. This information is added in the revised manuscript. The fused silica rod was a commercial product and no information on the position of this rod in the ingot is available.

  1. Part of the absorbed laser energy will be converted to fluorescence radiation, which is not detected by the laser calorimetric method. It would be interesting to have an estimation concerning the fraction of this loss channel compared to the absorption value detected by the laser calorimeter.

Reply: To our knowledge, there is no report on the fraction of the absorbed energy lost via radiative relaxation (photoluminescence). The common knowledge is that this fraction is small as compared to that via non-radiative relaxation (heat) detected by the laser calorimeter. This point is mentioned in the revised manuscript.

  1. The author may discuss and assess effects of nonlinear surface absorption.

Reply: Previous experimental results demonstrated that the surface absorption showed a linear effect. This point is mentioned and a reference is cited in the revised manuscript.
